# IMPLICIT MAXIMUM LIKELIHOOD ESTIMATION

## ABSTRACT

Implicit probabilistic models are models defined naturally in terms of a sampling procedure and often induces a likelihood function that cannot be expressed explicitly. We develop a simple method for estimating parameters in implicit models that does not require knowledge of the form of the likelihood function or any derived quantities, but can be shown to be equivalent to maximizing likelihood under some conditions. Our result holds in the non-asymptotic parametric setting, where both the capacity of the model and the number of data examples are finite. We also demonstrate encouraging experimental results.

## 1 INTRODUCTION

Generative modelling is a cornerstone of machine learning and has received increasing attention. Recent models like variational autoencoders (VAEs) (Kingma & Welling, 2013; Rezende et al., 2014) and generative adversarial nets (GANs) (Goodfellow et al., 2014; Gutmann et al., 2014), have delivered impressive advances in performance and generated a lot of excitement.

Generative models can be classified into two categories: *prescribed models* and *implicit models* (Diggle & Gratton, 1984; Mohamed & Lakshminarayanan, 2016). Prescribed models are defined by an explicit specification of the density, and so their unnormalized complete likelihood can be usually expressed in closed form. Examples include models whose complete likelihoods lie in the exponential family, such as mixture of Gaussians (Everitt, 1985), hidden Markov models (Baum & Petrie, 1966), Boltzmann machines (Hinton & Sejnowski, 1986). Because computing the normalization constant, also known as the partition function, is generally intractable, sampling from these models is challenging.

On the other hand, implicit models are defined most naturally in terms of a (simple) sampling procedure. Most models take the form of a deterministic parameterized transformation $T_\theta(\cdot)$ of an analytic distribution, like an isotropic Gaussian. This can be naturally viewed as the distribution induced by the following sampling procedure:

1. Sample $\mathbf{z} \sim \mathcal{N}(0, \mathbf{I})$
2. Return $\mathbf{x} := T_\theta(\mathbf{z})$

The transformation $T_\theta(\cdot)$ often takes the form of a highly expressive function approximator, like a neural net. Examples include generative adversarial nets (GANs) (Goodfellow et al., 2014; Gutmann et al., 2014) and generative moment matching nets (GMMNs) (Li et al., 2015; Dziugaite et al., 2015). The marginal likelihood of such models can be characterized as follows:

$$p_\theta(\mathbf{x}) = \frac{\partial}{\partial x_1} \cdots \frac{\partial}{\partial x_d} \int_{\left\{ \mathbf{z} \middle| \forall i \ (T_\theta(\mathbf{z}))_i \leq x_i \right\}} \phi(\mathbf{z}) d\mathbf{z}$$

where $\phi(\cdot)$ denotes the probability density function (PDF) of $\mathcal{N}(0, \mathbf{I})$.

In general, attempting to reduce this to a closed-form expression is hopeless. Evaluating it numerically is also challenging, since the domain of integration could consist of an exponential number of disjoint regions and numerical differentiation is ill-conditioned.

These two categories of generative models are not mutually exclusive. Some models admit both an explicit specification of the density and a simple sampling procedure and so can be considered as both prescribed and implicit. Examples include variational autoencoders (Kingma & Welling,

2013; Rezende et al., 2014), their predecessors (MacKay, 1995; Bishop et al., 1998) and extensions (Burda et al., 2015), and directed/autoregressive models, e.g., (Neal, 1992; Bengio & Bengio, 2000; Larochelle & Murray, 2011; van den Oord et al., 2016).

## 1.1 CHALLENGES IN PARAMETER ESTIMATION

Maximum likelihood (Fisher, 1912; Edgeworth, 1908) is perhaps the standard method for estimating the parameters of a probabilistic model from observations. The maximum likelihood estimator (MLE) has a number of appealing properties: under mild regularity conditions, it is asymptotically consistent, efficient and normal. A long-standing challenge of training probabilistic models is the computational roadblocks of maximizing the log-likelihood function directly.

For prescribed models, maximizing likelihood directly requires computing the partition function, which is intractable for all but the simplest models. Many powerful techniques have been developed to attack this problem, including variational methods (Jordan et al., 1999), contrastive divergence (Hinton, 2002; Welling & Hinton, 2002), score matching (Hyvärinen, 2005) and pseudolikelihood maximization (Besag, 1975), among others.

For implicit models, the situation is even worse, as there is no term in the log-likelihood function that is in closed form; evaluating any term requires computing an intractable integral. As a result, maximizing likelihood in this setting seems hopelessly difficult. A variety of likelihood-free solutions have been proposed that in effect minimize a divergence measure between the data distribution and the model distribution. They come in two forms: those that minimize an $f$-divergence, and those that minimize an integral probability metric (Müller, 1997). In the former category are GANs, which are based on the idea of minimizing the distinguishability between data and samples (Tu, 2007; Gutmann & Hyvärinen, 2010). It has been shown that when given access to an infinitely powerful discriminator, the original GAN objective minimizes the Jensen-Shannon divergence, the $-\log D$ variant of the objective minimizes the reverse KL-divergence minus a bounded quantity (Arjovsky & Bottou, 2017), and later extensions (Nowozin et al., 2016) minimize arbitrary $f$-divergences. In the latter category are GMMNs which use maximum mean discrepancy (MMD) (Gretton et al., 2007) as the witness function.

In the case of GANs, despite the theoretical results, there are a number of challenges that arise in practice, such as mode dropping/collapse (Goodfellow et al., 2014; Arora & Zhang, 2017), vanishing gradients (Arjovsky & Bottou, 2017; Sinn & Rawat, 2017) and training instability (Goodfellow et al., 2014; Arora et al., 2017). A number of explanations have been proposed to explain these phenomena and point out that many theoretical results rely on three assumptions: the discriminator must have infinite modelling capacity (Goodfellow et al., 2014; Arora et al., 2017), the number of samples from the true data distribution must be infinite (Arora et al., 2017; Sinn & Rawat, 2017) and the gradient ascent-descent procedure (Arrow et al., 1958; Schmidhuber, 1992) can converge to a global pure-strategy Nash equilibrium (Goodfellow et al., 2014; Arora et al., 2017). When some of these assumptions do not hold, the theoretical guarantees do not necessarily apply. A number of ways have been proposed that alleviate some of these issues, e.g., (Zhao et al., 2016; Salimans et al., 2016; Donahue et al., 2016; Dumoulin et al., 2016; Arjovsky et al., 2017; Hjelm et al., 2017; Li et al., 2017; Zhu et al., 2017), but a way of solving all three issues simultaneously remains elusive.

## 1.2 OUR CONTRIBUTION

In this paper, we present an alternative method for estimating parameters in implicit models. Like the methods above, our method is likelihood-free, but can be shown to be equivalent to maximizing likelihood under some conditions. Our result holds when the capacity of the model is finite and the number of data examples is finite. The idea behind the method is simple: it finds the nearest sample to each data example and optimizes the model parameters to pull the sample towards it. The direction in which nearest neighbour search is performed is important: the proposed method ensures each data example has a similar sample, which contrasts with an alternative approach of pushing each sample to the nearest data example, which would ensure that each sample has a similar data example. The latter approach would permit all samples being similar to one data example. Such a scenario would be heavily penalized by the former approach.

The proposed method could sidestep the three issues mentioned above: mode collapse, vanishing gradients and training instability. Modes are not dropped because the loss ensures each data example has a sample nearby at optimality; gradients do not vanish because the gradient of the distance between a data example and its nearest sample does not become zero unless they coincide; training is stable because the estimator is the solution to a simple minimization problem. By leveraging recent advances in fast nearest neighbour search algorithms (Li & Malik, 2016; 2017), this approach is able to scale to large, high-dimensional datasets.

## 2 IMPLICIT MAXIMUM LIKELIHOOD ESTIMATOR

### 2.1 DEFINITION

We are given a set of $n$ data examples $\mathbf{x}_1, \ldots, \mathbf{x}_n$ and some unknown parameterized probability distribution $P_\theta$ with density $p_\theta$. We also have access to an oracle that allows us to draw independent and identically distributed (i.i.d.) samples from $P_\theta$.

Let $\tilde{\mathbf{x}}_1^\theta, \ldots, \tilde{\mathbf{x}}_m^\theta$ be i.i.d. samples from $P_\theta$, where $m \geq n$. For each data example $\mathbf{x}_i$, we define a random variable $R_i^\theta$ to be the distance between $\mathbf{x}_i$ and the nearest sample. More precisely,

$$R_i^\theta := \min_{j \in [m]} \left\| \tilde{\mathbf{x}}_j^\theta - \mathbf{x}_i \right\|_2^2$$

where $[m]$ denotes $\{1, \ldots, m\}$.

The implicit maximum likelihood estimator $\hat{\theta}_{\text{IMLE}}$ is defined as:

$$\hat{\theta}_{\text{IMLE}} := \arg\min_\theta \mathbb{E}_{R_1^\theta, \ldots, R_n^\theta} \left[ \sum_{i=1}^n R_i^\theta \right]$$

$$= \arg\min_\theta \mathbb{E}_{\tilde{\mathbf{x}}_1^\theta, \ldots, \tilde{\mathbf{x}}_m^\theta} \left[ \sum_{i=1}^n \min_{j \in [m]} \left\| \tilde{\mathbf{x}}_j^\theta - \mathbf{x}_i \right\|_2^2 \right]$$

### 2.2 ALGORITHM

We outline the proposed parameter estimation procedure in Algorithm 1. In each outer iteration, we draw $m$ i.i.d. samples from the current model $P_\theta$. We then randomly select a batch of examples from the dataset and find the nearest sample from each data example. We then run a standard iterative optimization algorithm, like stochastic gradient descent (SGD), to minimize a sample-based version of the Implicit Maximum Likelihood Estimator (IMLE) objective.

---

**Algorithm 1** Implicit maximum likelihood estimation (IMLE) procedure

---

**Require:** The dataset $D = \{\mathbf{x}_i\}_{i=1}^n$ and a sampling mechanism for the implicit model $P_\theta$
    Initialize $\theta$ to a random vector
    **for** $k = 1$ **to** $K$ **do**
        Draw i.i.d. samples $\tilde{\mathbf{x}}_1^\theta, \ldots, \tilde{\mathbf{x}}_m^\theta$ from $P_\theta$
        Pick a random batch $S \subseteq \{1, \ldots, n\}$
        $\sigma(i) \leftarrow \arg\min_j \left\| \mathbf{x}_i - \tilde{\mathbf{x}}_j^\theta \right\|_2^2 \ \forall i \in S$
        **for** $l = 1$ **to** $L$ **do**
            Pick a random mini-batch $\tilde{S} \subseteq S$
            $\theta \leftarrow \theta - \eta \nabla_\theta \left( \frac{n}{|\tilde{S}|} \sum_{i \in \tilde{S}} \left\| \mathbf{x}_i - \tilde{\mathbf{x}}_{\sigma(i)}^\theta \right\|_2^2 \right)$
        **end for**
    **end for**
    **return** $\theta$

---

Because our algorithm needs to solve a nearest neighbour search problem in each outer iteration, the scalability of our method depends on our ability to find the nearest neighbours quickly. This was traditionally considered to be a hard problem, especially in high dimensions. However, this is no

longer the case, due to recent advances in nearest neighbour search algorithms (Li & Malik, 2016; 2017).

Note that the use of Euclidean distance is not a major limitation of the proposed approach. A variety of distance metrics are either exactly or approximately equivalent to Euclidean distance in some non-linear embedding space, in which case the theoretical guarantees are inherited from the Euclidean case. This encompasses popular distance metrics used in the literature, like the Euclidean distance between the activations of a neural net, which is often referred to as a perceptual similarity metric (Salimans et al., 2016; Dosovitskiy & Brox, 2016). The approach can be easily extended to use these metrics, though because this is the initial paper on this method, we focus on the vanilla setting of Euclidean distance in the natural representation of the data, e.g.: pixels, both for simplicity/clarity and for comparability to vanilla versions of other methods that do not use auxiliary sources of labelled data or leverage domain-specific prior knowledge. For distance metrics that cannot be embedded in Euclidean space, the analysis can be easily adapted with minor modifications as long as the volume of a ball under the metric has a simple dependence on its radius.

## 3    WHY MAXIMUM LIKELIHOOD

There has been debate (Huszár, 2015) over whether maximizing likelihood of the data is the appropriate objective for the purposes of learning generative models. Recall that maximizing likelihood is equivalent to minimizing $D_{KL}(p_{\text{data}} \| p_\theta)$, where $p_{\text{data}}$ denotes the empirical data distribution and $p_\theta$ denotes the model distribution. One proposed alternative is to minimize the reverse KL-divergence, $D_{KL}(p_\theta \| p_{\text{data}})$, which is suggested (Huszár, 2015) to be better because it severely penalizes the model for generating an implausible sample, whereas the standard KL-divergence, $D_{KL}(p_{\text{data}} \| p_\theta)$, severely penalizes the model for assigning low density to a data example. As a result, when the model is underspecified, i.e. has less capacity than what's necessary to fit all the modes of the data distribution, minimizing $D_{KL}(p_\theta \| p_{\text{data}})$ leads to a narrow model distribution that concentrates around a few modes, whereas minimizing $D_{KL}(p_{\text{data}} \| p_\theta)$ leads to a broad model distribution that hedges between modes. The success of GANs in generating good samples is often attributed to the former phenomenon (Arjovsky & Bottou, 2017).

This argument, however, relies on the assumption that we have access to an infinite number of samples from the true data distribution. In practice, however, this assumption rarely holds: if we had access to the true data distribution, then there is usually no need to fit a generative model, since we can simply draw samples from the true data distribution. What happens when we only have the empirical data distribution? Recall that $D_{KL}(p \| q)$ is defined and finite only if $p$ is absolutely continuous w.r.t. $q$, i.e.: $q(x) = 0$ implies $p(x) = 0$ for all $x$. In other words, $D_{KL}(p \| q)$ is defined and finite only if the support of $p$ is contained in the support of $q$. Now, consider the difference between $D_{KL}(p_{\text{data}} \| p_\theta)$ and $D_{KL}(p_\theta \| p_{\text{data}})$: minimizing the former, which is equivalent to maximizing likelihood, ensures that the support of the model distribution contains all data examples, whereas minimizing the latter ensures that the support of the model distribution is contained in the support of the empirical data distribution, which is just the set of data examples. In other words, maximum likelihood disallows mode dropping, whereas minimizing reverse KL-divergence forces the model to assign zero density to unseen data examples and effectively prohibits generalization. Furthermore, maximum likelihood discourages the model from assigning low density to any data example, since doing so would make the likelihood, which is the product of the densities at each of the data examples, small.

From the modelling perspective, because maximum likelihood is guaranteed to preserve all modes, it can make use of all available training data and can therefore be used to train high-capacity models that have a large number of parameters. In contrast, using an objective that permits mode dropping allows the model to pick and choose which data examples it wants to model. As a result, if the goal is to train a high-capacity model that can learn the underlying data distribution, we would not be able to do so using such an objective because we have no control over which modes the model chooses to drop. Put another way, we can think about the model's performance along two axes: its ability to generate plausible samples (precision) and its ability to generate all modes of the data distribution (recall). A model that successfully learns the underlying distribution should score high along both axes. If mode dropping is allowed, then an improvement in precision may be achieved at the expense of lower recall and could represent a move to a different point on the same precision-recall

curve. As a result, since sample quality is an indicator of precision, improvement in sample quality in this setting may not mean an improvement in density estimation performance. On the other hand, if mode dropping is disallowed, since full recall is always guaranteed, an improvement in precision is achieved without sacrificing recall and so implies an upwards shift in the precision-recall curve. In this case, an improvement in sample quality does signify an improvement in density estimation performance, which may explain sample quality historically was an important way to evaluate the performance of generative models, most of which maximized likelihood. With the advent of generative models that permit mode dropping, however, sample quality is no longer a reliable indicator of density estimation performance, since good sample quality can be trivially achieved by dropping all but a few modes. In this setting, sample quality can be misleading, since a model with low recall on a lower precision-recall curve can achieve a better precision than a model with high recall on a higher precision-recall curve. Since it is hard to distinguish whether an improvement in sample quality is due to a move along the same precision-recall curve or a real shift in the curve, an objective that disallows mode dropping is critical tool that researchers can use to develop better models, since they can be sure that an apparent improvement in sample quality is due to a shift in the precision-recall curve.

## 4 ANALYSIS

Before formally stating the theoretical results, we first illustrate the intuition behind why the proposed estimator is equivalent to maximum likelihood estimator under some conditions. For simplicity, we will consider the special case where we only have a single data example $\mathbf{x}_1$ and a single sample $\tilde{\mathbf{x}}_1^\theta$. Consider the total density of $P_\theta$ inside a ball of radius of $t$ centred at $\mathbf{x}_1$ as a function of $t$, a function that will be denoted as $\tilde{F}^\theta(t)$. If the density in the neighbourhood of $\mathbf{x}_1$ is high, then $\tilde{F}^\theta(t)$ would grow rapidly as $t$ increases. If, on the other hand, the density in the neighbourhood of $\mathbf{x}_1$ is low, then $\tilde{F}^\theta(t)$ would grow slowly. So, maximizing likelihood is equivalent to making $\tilde{F}^\theta(t)$ grow as fast as possible. To this end, we can maximize the area under the function $\tilde{F}^\theta(t)$, or equivalently, minimize the area under the function $1 - \tilde{F}^\theta(t)$. Observe that $\tilde{F}^\theta(t)$ can be interpreted as the cumulative distribution function (CDF) of the Euclidean distance between $\mathbf{x}_1$ and $\tilde{\mathbf{x}}_1^\theta$, which is a random variable because $\tilde{\mathbf{x}}_1^\theta$ is random and will be denoted as $\tilde{R}^\theta$. Because $\tilde{R}^\theta$ is non-negative, recall that $\mathbb{E}\left[\tilde{R}^\theta\right] = \int_0^\infty \Pr\left(\tilde{R}^\theta > t\right) dt = \int_0^\infty \left(1 - \tilde{F}^\theta(t)\right) dt$, which is exactly the area under the function $1 - \tilde{F}^\theta(t)$. Therefore, we can maximize likelihood of a data example $\mathbf{x}_1$ by minimizing $\mathbb{E}\left[\tilde{R}^\theta\right]$, or in other words, minimizing the expected distance between the data example and a random sample. To extend this analysis to the case with multiple data examples, we show in the supplementary material that if the objective function is a summation, applying a monotonic transformation to each term and then reweighting appropriately preserves the optimizer under some conditions.

We now state the key theoretical result formally. Please refer to the supplementary material for the proof.

**Theorem 1.** *Consider a set of observations* $\mathbf{x}_1, \ldots, \mathbf{x}_n$, *a parameterized family of distributions* $P_\theta$ *with probability density function (PDF)* $p_\theta(\cdot)$ *and a unique maximum likelihood solution* $\theta^*$. *For any* $m \geq 1$, *let* $\tilde{\mathbf{x}}_1^\theta, \ldots, \tilde{\mathbf{x}}_m^\theta \sim P_\theta$ *be i.i.d. random variables and define* $\tilde{r}^\theta := \left\|\tilde{\mathbf{x}}_1^\theta\right\|_2^2$, $R^\theta := \min_{j \in [m]} \left\|\tilde{\mathbf{x}}_j^\theta\right\|_2^2$ *and* $R_i^\theta := \min_{j \in [m]} \left\|\tilde{\mathbf{x}}_j^\theta - \mathbf{x}_i\right\|_2^2$. *Let* $F^\theta(\cdot)$ *be the cumulative distribution function (CDF) of* $\tilde{r}^\theta$ *and* $\Psi(z) := \min_\theta \left\{\mathbb{E}\left[R^\theta\right] | p_\theta(\mathbf{0}) = z\right\}$.

*If* $P_\theta$ *satisfies the following:*

- $p_\theta(\mathbf{x})$ *is differentiable w.r.t.* $\theta$ *and continuous w.r.t.* $\mathbf{x}$ *everywhere.*

- $\forall \theta, \mathbf{v}$, *there exists* $\theta'$ *such that* $p_\theta(\mathbf{x}) = p_{\theta'}(\mathbf{x} + \mathbf{v}) \; \forall \mathbf{x}$.

- *For any* $\theta_1, \theta_2$, *there exists* $\theta_0$ *such that* $F^{\theta_0}(t) \geq \max\left\{F^{\theta_1}(t), F^{\theta_2}(t)\right\} \; \forall t \geq 0$ *and* $p_{\theta_0}(\mathbf{0}) = \max\left\{p_{\theta_1}(\mathbf{0}), p_{\theta_2}(\mathbf{0})\right\}$.

| Method | MNIST | TFD |
|---|---|---|
| DBN (Bengio et al., 2013) | $138 \pm 2$ | $1909 \pm 66$ |
| SCAE (Bengio et al., 2013) | $121 \pm 1.6$ | $2110 \pm 50$ |
| DGSN (Bengio et al., 2014) | $214 \pm 1.1$ | $1890 \pm 29$ |
| GAN (Goodfellow et al., 2014) | $225 \pm 2$ | $2057 \pm 26$ |
| GMMN (Li et al., 2015) | $147 \pm 2$ | $2085 \pm 25$ |
| IMLE (Proposed) | $\mathbf{257 \pm 6}$ | $\mathbf{2139 \pm 27}$ |

Table 1: Log-likelihood of the test data under the Gaussian Parzen window density estimated from samples generated by different methods.

- *$\exists \tau > 0$ such that $\forall i \in [n]\ \forall \theta \notin B_{\theta^*}(\tau)$, $p_\theta(\mathbf{x}_i) < p_{\theta^*}(\mathbf{x}_i)$, where $B_{\theta^*}(\tau)$ denotes the ball centred at $\theta^*$ of radius $\tau$.*

- *$\Psi(z)$ is differentiable everywhere.*

- *For all $\theta$, if $\theta \neq \theta^*$, there exists $j \in [d]$ such that*
$$\left\langle \begin{pmatrix} \frac{\Psi'(p_\theta(\mathbf{x}_1))p_\theta(\mathbf{x}_1)}{\Psi'(p_{\theta^*}(\mathbf{x}_1))p_{\theta^*}(\mathbf{x}_1)} \\ \vdots \\ \frac{\Psi'(p_\theta(\mathbf{x}_n))p_\theta(\mathbf{x}_n)}{\Psi'(p_{\theta^*}(\mathbf{x}_n))p_{\theta^*}(\mathbf{x}_n)} \end{pmatrix}, \begin{pmatrix} \nabla_\theta \left(\log p_\theta(\mathbf{x}_1)\right)_j \\ \vdots \\ \nabla_\theta \left(\log p_\theta(\mathbf{x}_n)\right)_j \end{pmatrix} \right\rangle \neq 0.$$

*Then,*

$$\arg\min_\theta \sum_{i=1}^n \frac{\mathbb{E}\left[R_i^\theta\right]}{\Psi'(p_{\theta^*}(\mathbf{x}_i))p_{\theta^*}(\mathbf{x}_i)} = \arg\max_\theta \sum_{i=1}^n \log p_\theta(\mathbf{x}_i)$$

*Furthermore, if $p_{\theta^*}(\mathbf{x}_1) = \cdots = p_{\theta^*}(\mathbf{x}_n)$, then,*

$$\arg\min_\theta \sum_{i=1}^n \mathbb{E}\left[R_i^\theta\right] = \arg\max_\theta \sum_{i=1}^n \log p_\theta(\mathbf{x}_i)$$

Now, we examine the restrictiveness of each condition. The first condition is satisfied by nearly all analytic distributions. The second condition is satisfied by nearly all distributions that have an unrestricted location parameter, since one can simply shift the location parameter by $\mathbf{v}$. The third condition is satisfied by most distributions that have location and scale parameters, like a Gaussian distribution, since the scale can be made arbitrarily low and the location can be shifted so that the constraint on $p_\theta(\cdot)$ is satisfied. The fourth condition is satisfied by nearly all distributions, whose density eventually tends to zero as the distance from the optimal parameter setting tends to infinity. The fifth condition requires $\min_\theta \left\{ \mathbb{E}\left[R^\theta\right] | p_\theta(\mathbf{0}) = z \right\}$ to change smoothly as $\mathbf{z}$ changes. The final condition requires the two $n$-dimensional vectors, one of which can be chosen from a set of $d$ vectors, to be not exactly orthogonal. As a result, this condition is usually satisfied when $d$ is large, i.e. when the model is richly parameterized.

There is one remaining difficulty in applying this theorem, which is that the quantity $1/\Psi'(p_{\theta^*}(\mathbf{x}_i))p_{\theta^*}(\mathbf{x}_i)$, which appears as an coefficient on each term in the proposed objective, is typically not known. If we consider a new objective that ignores the coefficients, i.e. $\sum_{i=1}^n \mathbb{E}\left[R_i^\theta\right]$, then minimizing this objective is equivalent to minimizing an upper bound on the ideal objective, $\sum_{i=1}^n \mathbb{E}\left[R_i^\theta\right]/\Psi'(p_{\theta^*}(\mathbf{x}_i))p_{\theta^*}(\mathbf{x}_i)$. The tightness of this bound depends on the difference between the highest and lowest likelihood assigned to individual data points at the optimum, i.e. the maximum likelihood estimate of the parameters. Such a model should not assign high likelihoods to some points and low likelihoods to others as long as it has reasonable capacity, since doing so would make the overall likelihood, which is the product of the likelihoods of individual data points, low. Therefore, the upper bound is usually reasonably tight.

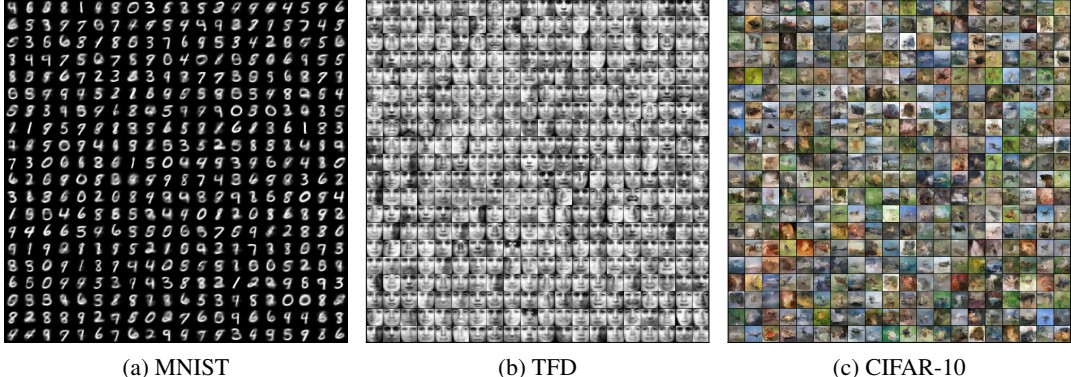

 (b) TFD (c) CIFAR-10

Figure 1: Representative random samples from the model trained on (a) MNIST, (b) Toronto Faces Dataset and (c) CIFAR-10.

## 5 EXPERIMENTS

We trained generative models using the proposed method on three standard benchmark datasets, MNIST, the Toronto Faces Dataset (TFD) and CIFAR-10. All models take the form of feedforward neural nets with isotropic Gaussian noise as input.

For MNIST, the architecture consists of two fully connected hidden layers with 1200 units each followed by a fully connected output layer with 784 units. ReLU activations were used for hidden layers and sigmoids were used for the output layer. For TFD, the architecture is wider and consists of two fully connected hidden layers with 8000 units each followed by a fully connected output layer with 2304 units. For both MNIST and TFD, the dimensionality of the noise vector is 100.

For CIFAR-10, we used a simple convolutional architecture with 1000-dimensional Gaussian noise as input. The architecture consists of five convolutional layers with 512 output channels and a kernel size of 5 that all produce $4 \times 4$ feature maps, followed by a bilinear upsampling layer that doubles the width and height of the feature maps. There is a batch normalization layer followed by leaky ReLU activations with slope $-0.2$ after each convolutional layer. This design is then repeated for each subsequent level of resolution, namely $8 \times 8$, $16 \times 16$ and $32 \times 32$, so that we have 20 convolutional layers, each with output 512 channels. We then add a final output layer with three output channels on top, followed by sigmoid activations. We note that this architecture has more capacity than typical architectures used in other methods, like (Radford et al., 2015). This is because our method aims to capture all modes of the data distribution and therefore needs more modelling capacity than methods that are permitted to drop modes.

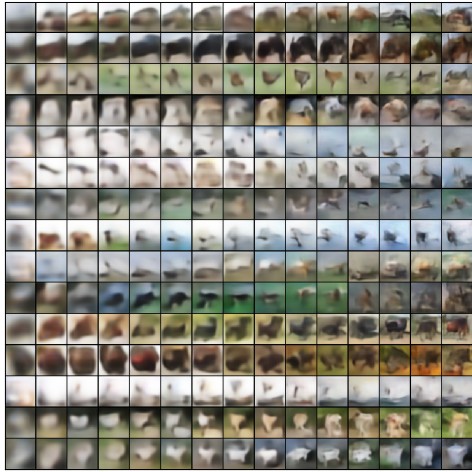

Figure 2: Samples corresponding to the same latent variable values at different points in time while training the model on CIFAR-10. Each row corresponds to a sample, and each column corresponds to a particular point in time.

Evaluation for implicit generative models in general remains an open problem. Various intrinsic and extrinsic evaluation metrics have been proposed, all of which have limitations. Extrinsic evaluation metrics measure performance indirectly via a downstream task (Salimans et al., 2016). Unfortunately, dependence on the downstream task could introduce bias and may not capture de-

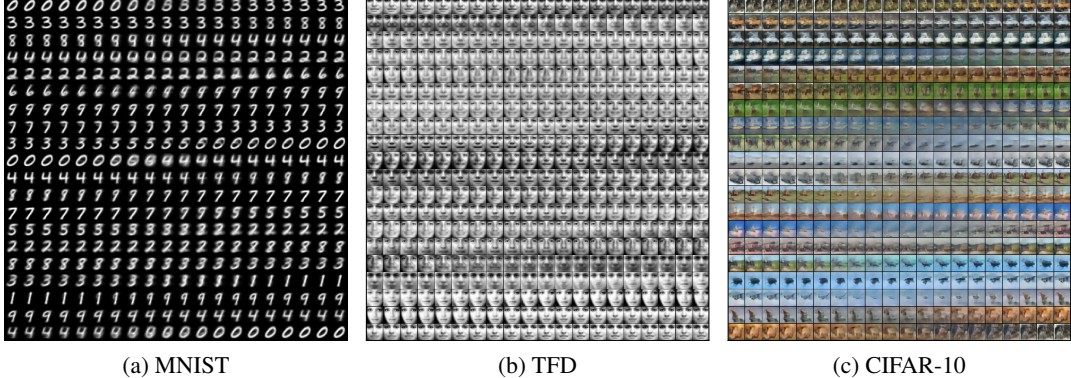

|       (a) MNIST       |        (b) TFD        |      (c) CIFAR-10      |

Figure 3: Linear interpolation between samples in code space. The first image in every row is an independent sample; all other images are interpolated between the previous and the subsequent sample. Images along the path of interpolation are shown in the figure arranged from from left to right, top to bottom. They also wrap around, so that images in the last row are interpolations between the last and first samples.

sirable properties of the generative model that do not affect performance on the task. Intrinsic evaluation metrics measure performance without relying on external models or data. Popular examples include estimated log-likelihood (Bengio et al., 2014; Wu et al., 2016) and visual assessment of sample quality. While recent literature has focused more on the latter and less on the former, it should be noted that they evaluate different properties – sample quality reflects precision, i.e.: how accurate the model samples are compared to the ground truth, whereas estimated log-likelihood focuses on recall, i.e.: how much of the diversity in the data distribution the model captures. Consequently, both are important metrics; one is not a replacement for the other. As pointed out by (Theis et al., 2015), "qualitative as well as quantitative analyses based on model samples can be misleading about a model's density estimation performance, as well as the probabilistic model's performance in applications other than image synthesis." Two models that achieve different levels of precision may simply be at different points on the same precision-recall curve, and therefore may not be directly comparable. Models that achieve the same level of recall, on the other hand, may be directly compared. So, for methods that maximize likelihood, which are guaranteed to preserve all modes and achieve full recall, both sample quality and estimated log-likelihood capture precision. Because most generative models traditionally maximized likelihood or a lower bound on the likelihood, the only property that differed across models was precision, which may explain why sample quality has historically been seen as an important indicator of performance. However, in heterogenous experimental settings with different models optimized for various objectives, sample quality does not necessarily reflect how well a model learns the underlying data distribution. Therefore, under these settings, both precision and recall need to be measured. While there is not yet a reliable way to measure recall (given the known issues of estimated log-likelihoods in high dimensions), this does not mean that sample quality can be a valid substitute for estimated log-likelihoods, as it cannot detect the lack of diversity of samples. A secondary issue that is more easily solvable is that samples presented in papers are sometimes cherry-picked; as a result, they capture the maximum sample quality, but not necessarily the mean sample quality.

To mitigate these problems to some extent, we avoid cherry-picking and visualize randomly chosen samples, which are shown in Figure 1. We also report the estimated log-likelihood in Table 1. As mentioned above, both evaluation criteria have biases/deficiencies, so performing well on either of these metrics does not necessarily indicate good density estimation performance. However, not performing badly on either metric can provide some comfort that the model is simultaneously able to achieve reasonable precision and recall.

As shown in Figure 1, despite its simplicity, the proposed method is able to generate reasonably good samples for MNIST, TFD and CIFAR-10. While it is commonly believed that minimizing reverse KL-divergence is necessary to produce good samples and maximizing likelihood necessarily leads to poor samples (Grover et al., 2017), the results suggest that this is not necessarily the case. Even

though Euclidean distance was used in the objective, the samples do not appear to be desaturated or overly blurry. Samples also seem fairly diverse. This is supported by the estimated log-likelihood results in Table 1. Because the model achieved a high score on that metric on both MNIST and TFD, this suggests that the model did not suffer from significant mode dropping.

In Figure 4 in the supplementary material, we show samples and their nearest neighbours in the training set. Each sample is quite different from its nearest neighbour in the training set, suggesting that the model has not overfitted to examples in the training set.

Next, we visualize the learned manifold by walking along a geodesic on the manifold between pairs of samples. More concretely, we generate five samples, arrange them in arbitrary order, perform linear interpolation in latent variable space between adjacent pairs of samples, and generate an image from the interpolated latent variable. As shown in Figure 3, the images along the path of interpolation appear visually plausible and do not have noisy artifacts. In addition, the transition from one image to the next appears smooth, including for CIFAR-10, which contrasts with findings in the literature that suggest the transition between two natural images tends to be abrupt. This indicates that the support of the model distribution has not collapsed to a set of isolated points and that the proposed method is able to learn the geometry of the data manifold, even though it does not learn a distance metric explicitly.

Finally, we illustrate the evolution of samples as training progresses in Figure 2. As shown, the samples are initially blurry and become sharper over time. Importantly, sample quality consistently improves over time, which demonstrates the stability of training.

While our sample quality may not be state-of-the-art, it is important to remember that these results are obtained under the setting of full recall. So, this does not necessarily mean that our method models the underlying data distribution less accurately than other methods that achieve better sample quality, as some of them may drop modes and therefore achieve less than full recall. As previously mentioned, this does not suggest a fundamental tradeoff between precision and recall that cannot be overcome – on the contrary, our method provides researchers with a way of designing models that can improve the precision-recall curve without needing to worry that the observed improvements are due to a movement along the curve. With refinements to the model, it is possible to move the curve upwards and obtain better sample quality at any level of recall as a consequence. This is left for future work; as this is the initial paper on this approach, its value stems from the foundation it lays for a new research direction upon which subsequent work can be built, as opposed to the current results themselves. For this paper, we made a deliberate decision to keep the model simple, since non-essential practically motivated enhancements are less grounded in theory, may obfuscate the key underlying idea and could impart the impression that they are critical to making the approach work in practice. The fact that our method is able to generate more plausible samples on CIFAR-10 than other methods at similar stages of development, such as the initial versions of GAN (Goodfellow et al., 2014) and PixelRNN (van den Oord et al., 2016), despite the minimal sophistication of our method and architecture, shows the promise of the approach. Later iterations of other methods incorporate additional supervision in the form of pretrained weights and/or make task-specific modifications to the architecture and training procedure, which were critical to achieving state-of-the-art sample quality. We do believe the question of how the architecture should be refined in the context of our method to take advantage of task-specific insights is an important one, and is an area ripe for future exploration.

## 6 DISCUSSION

In this section, we consider and address some possible concerns about our method.

### 6.1 DOES MAXIMIZING LIKELIHOOD NECESSARILY LEAD TO POOR SAMPLE QUALITY?

It has been suggested (Huszár, 2015) that maximizing likelihood leads to poor sample quality because when the model is underspecified, it will try to cover all modes of the empirical data distribution and therefore assign high density to regions with few data examples. There is also empirical evidence (Grover et al., 2017) for a negative correlation between sample quality and log likelihood, suggesting an inherent trade-off between maximizing likelihood and achieving good sample quality. A popular solution is to minimize reverse KL-divergence instead, which trades off recall for pre-

cision. This is an imperfect solution, as the ultimate goal is to model all the modes *and* generate high-quality samples.

Note that this apparent trade-off exists that the model capacity is assumed to be fixed. We argue that a more promising approach would be to increase the capacity of the model, so that it is less underspecified. As the model capacity increases, avoiding mode dropping becomes more important, because otherwise there will not be enough training data to fit the larger number of parameters to. This is precisely a setting appropriate for maximum likelihood. As a result, it is possible that a combination of increasing the model capacity and maximum likelihood training can achieve good precision and recall simultaneously.

### 6.2 Would Minimizing Distance to the Nearest Samples Cause Overfitting?

When the model has infinite capacity, minimizing distance from data examples to their nearest samples will lead to a model distribution that memorizes data examples. The same is true if we maximize likelihood. Likewise, minimizing any divergence measure will lead to memorization of data examples, since the minimum divergence is zero and by definition, this can only happen if the model distribution is the same as the *empirical* data distribution, whose support is confined to the set of data examples. This implies that whenever we have a finite number of data examples, any method that learns a model with infinite capacity will memorize the data examples and will hence overfit.

To get around this, most methods learn a parametric model with finite capacity. In the parametric setting, the minimum divergence is not necessarily zero; the same is true for the minimum distance from data examples to their nearest samples. Therefore, the optimum of these objective functions is not necessarily a model distribution that memorizes data examples, and so overfitting will not necessarily occur.

### 6.3 Does Disjoint Support Break Maximum Likelihood?

Arjovsky et al. (2017) observes that the data distribution and the model distribution are supported on low-dimensional manifolds and so they are unlikely to have a non-negligible intersection. They point out $D_{KL}(p_{\text{data}} \| p_\theta)$ would be infinite in this case, or equivalently, the likelihood would be zero. While this does not invalidate the theoretical soundness of maximum likelihood, since the maximum of a non-negative function that is zero almost everywhere is still well-defined, it does cause a lot of practical issues for gradient-based learning, as the gradient is zero almost everywhere. This is believed to be one reason that models like variational autoencoders (Kingma & Welling, 2013; Rezende et al., 2014) use a Gaussian distribution with high variance for the conditional likelihood/observation model rather than a distribution close to the Dirac delta, so that the support of the model distribution is broadened to cover all the data examples (Arjovsky et al., 2017).

This issue does not affect our method, as our loss function is different from the log-likelihood function, even though their optima are the same (under some conditions). As the result, the gradients of our loss function are different from those of log-likelihood. When the supports of the data distribution and the model distribution do not overlap, each data example is likely far away from its nearest sample and so the gradient is large. Moreover, the farther the data examples are from the samples, the larger the gradient gets. Therefore, even when the gradient of log-likelihood can be tractably computed, there may be situations when the proposed method would work better than maximizing likelihood directly.

## 7 Conclusion

We presented a simple and versatile method for parameter estimation when the form of the likelihood is unknown. The method works by drawing samples from the model, finding the nearest sample to every data example and adjusting the parameters of the model so that it is closer to the data example. We showed that performing this procedure is equivalent to maximizing likelihood under some conditions. The proposed method can capture the full diversity of the data and avoids common issues like mode collapse, vanishing gradients and training instability. The method combined with vanilla model architectures is able to achieve encouraging results on MNIST, TFD and CIFAR-10.

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

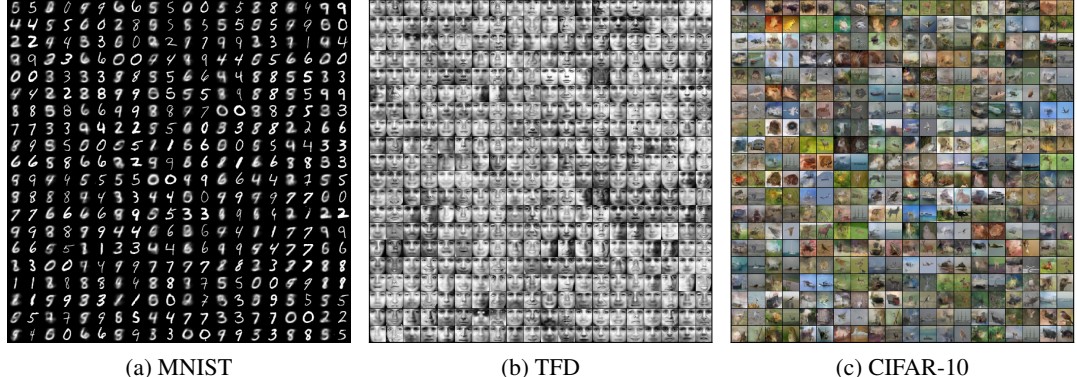

| (a) MNIST | (b) TFD | (c) CIFAR-10 |

Figure 4: Comparison of samples and their nearest neighbours in the training set. Images in odd-numbered columns are samples; to the right of each sample is its nearest neighbour in the training set.

## 8 SUPPLEMENTARY MATERIAL

Before proving the main result, we first prove the following intermediate results:

**Lemma 1.** *Let $\Omega \subseteq \mathbb{R}^d$ and $V \subseteq \mathbb{R}$. For $i \in [N]$, let $f_i : \Omega \to V$ be differentiable on $\Omega$ and $\Phi : V \to \mathbb{R}$ be differentiable on $V$ and strictly increasing. Assume $\arg\min_{\theta \in \Omega} \sum_{i=1}^N f_i(\theta)$ exists and is unique. Let $\theta^* := \arg\min_{\theta \in \Omega} \sum_{i=1}^N f_i(\theta)$ and $w_i := 1/\Phi'(f_i(\theta^*))$. If the following conditions hold:*

- *There is a bounded set $S \subseteq \Omega$ such that $\mathrm{bd}(S) \subseteq \Omega$, $\theta^* \in S$ and $\forall f_i$, $\forall \theta \in \Omega \setminus S$, $f_i(\theta) > f_i(\theta^*)$, where $\mathrm{bd}(S)$ denotes the boundary of $S$.*

- *For all $\theta \in \Omega$, if $\theta \neq \theta^*$, there exists $j \in [d]$ such that*
$$\left\langle \begin{pmatrix} w_1 \Phi'(f_1(\theta)) \\ \vdots \\ w_n \Phi'(f_n(\theta)) \end{pmatrix}, \begin{pmatrix} \partial f_1/\partial\theta_j(\theta) \\ \vdots \\ \partial f_n/\partial\theta_j(\theta) \end{pmatrix} \right\rangle \neq 0.$$

*Then $\arg\min_{\mathbf{x} \in \Omega} \sum_{i=1}^N w_i \Phi(f_i(\theta))$ exists and is unique. Furthermore, $\arg\min_{\theta \in \Omega} \sum_{i=1}^N w_i \Phi(f_i(\theta)) = \arg\min_{\theta \in \Omega} \sum_{i=1}^N f_i(\theta)$.*

*Proof.* Let $S \subseteq \Omega$ be the bounded set such that $\mathrm{bd}(S) \subseteq \Omega$, $\theta^* \in S$ and $\forall f_i$, $\forall \theta \in \Omega \setminus S$, $f_i(\theta) > f_i(\theta^*)$. Consider the closure of $S := S \cup \mathrm{bd}(S)$, denoted as $\bar{S}$. Because $S \subseteq \Omega$ and $\mathrm{bd}(S) \subseteq \Omega$, $\bar{S} \subseteq \Omega$. Since $S$ is bounded, $\bar{S}$ is bounded. Because $\bar{S} \subseteq \Omega \subseteq \mathbb{R}^d$ and is closed and bounded, it is compact.

Consider the function $\sum_{i=1}^N w_i \Phi(f_i(\cdot))$. By the differentiability of $f_i$'s and $\Phi$, $\sum_{i=1}^N w_i \Phi(f_i(\cdot))$ is differentiable on $\Omega$ and hence continuous on $\Omega$. By the compactness of $\bar{S}$ and the continuity of $\sum_{i=1}^N w_i \Phi(f_i(\cdot))$ on $\bar{S} \subseteq \Omega$, Extreme Value Theorem applies, which implies that $\min_{\theta \in \bar{S}} \sum_{i=1}^N w_i \Phi(f_i(\theta))$ exists. Let $\tilde{\theta} \in \bar{S}$ be such that $\sum_{i=1}^N w_i \Phi(f_i(\tilde{\theta})) = \min_{\theta \in \bar{S}} \sum_{i=1}^N w_i \Phi(f_i(\theta))$.

By definition of $S$, $\forall f_i$, $\forall \theta \in \Omega \setminus S$, $f_i(\theta) > f_i(\theta^*)$, implying that $\Phi(f_i(\theta)) > \Phi(f_i(\theta^*))$ since $\Phi$ is strictly increasing. Because $\Phi'(\cdot) > 0$, $w_i > 0$ and so $\sum_{i=1}^N w_i \Phi(f_i(\theta)) > \sum_{i=1}^N w_i \Phi(f_i(\theta^*))$ $\forall \theta \in \Omega \setminus S$. At the same time, since $\theta^* \in S \subset \bar{S}$, by definition of $\tilde{\theta}$, $\sum_{i=1}^N w_i \Phi(f_i(\tilde{\theta})) \leq \sum_{i=1}^N w_i \Phi(f_i(\theta^*))$. Combining these two facts yields $\sum_{i=1}^N w_i \Phi(f_i(\tilde{\theta})) \leq \sum_{i=1}^N w_i \Phi(f_i(\theta^*)) < \sum_{i=1}^N w_i \Phi(f_i(\theta))$ $\forall \theta \in \Omega \setminus S$. Since the inequality is strict, this implies that $\tilde{\theta} \notin \Omega \setminus S$, and so $\tilde{\theta} \in \bar{S} \setminus (\Omega \setminus S) \subseteq \Omega \setminus (\Omega \setminus S) = S$.

In addition, because $\tilde{\theta}$ is the minimizer of $\sum_{i=1}^{N} w_i \Phi(f_i(\cdot))$ on $\bar{S}$, $\sum_{i=1}^{N} w_i \Phi(f_i(\tilde{\theta})) \leq \sum_{i=1}^{N} w_i \Phi(f_i(\theta))$ $\forall \theta \in \bar{S}$. So, $\sum_{i=1}^{N} w_i \Phi(f_i(\tilde{\theta})) \leq \sum_{i=1}^{N} w_i \Phi(f_i(\theta))$ $\forall \theta \in \bar{S} \cup (\Omega \setminus S) \supseteq S \cup (\Omega \setminus S) = \Omega$. Hence, $\tilde{\theta}$ is a minimizer of $\sum_{i=1}^{N} w_i \Phi(f_i(\cdot))$ on $\Omega$, and so $\min_{\theta \in \Omega} \sum_{i=1}^{N} w_i \Phi(f_i(\theta))$ exists. Because $\sum_{i=1}^{N} w_i \Phi(f_i(\cdot))$ is differentiable on $\Omega$, $\tilde{\theta}$ must be a critical point of $\sum_{i=1}^{N} w_i \Phi(f_i(\cdot))$ on $\Omega$.

On the other hand, since $\Phi$ is differentiable on $V$ and $f_i(\theta) \in V$ for all $\theta \in \Omega$, $\Phi'(f_i(\theta))$ exists for all $\theta \in \Omega$. So,

$$\nabla \left( \sum_{i=1}^{N} w_i \Phi(f_i(\theta)) \right) = \sum_{i=1}^{N} w_i \nabla \left( \Phi(f_i(\theta)) \right)$$
$$= \sum_{i=1}^{N} w_i \Phi'(f_i(\theta)) \nabla f_i(\theta)$$
$$= \sum_{i=1}^{N} \frac{\Phi'(f_i(\theta))}{\Phi'(f_i(\theta^*))} \nabla f_i(\theta)$$

At $\theta = \theta^*$,

$$\nabla \left( \sum_{i=1}^{N} w_i \Phi(f_i(\theta^*)) \right) = \sum_{i=1}^{N} \frac{\Phi'(f_i(\theta^*))}{\Phi'(f_i(\theta^*))} \nabla f_i(\theta^*)$$
$$= \sum_{i=1}^{N} \nabla f_i(\theta^*)$$

Since each $f_i$ is differentiable on $\Omega$, $\sum_{i=1}^{N} f_i$ is differentiable on $\Omega$. Combining this with the fact that $\theta^*$ is the minimizer of $\sum_{i=1}^{N} f_i$ on $\Omega$, it follows that $\nabla \left( \sum_{i=1}^{N} f_i(\theta^*) \right) = \sum_{i=1}^{N} \nabla f_i(\theta^*) = 0$. Hence, $\nabla \left( \sum_{i=1}^{N} w_i \Phi(f_i(\theta^*)) \right) = 0$ and so $\theta^*$ is a critical point of $\sum_{i=1}^{N} w_i \Phi(f_i(\cdot))$.

Because $\forall \theta \in \Omega$, if $\theta \neq \theta^*$, $\exists j \in [d]$ such that $\left\langle \begin{pmatrix} w_1 \Phi'(f_1(\theta)) \\ \vdots \\ w_n \Phi'(f_n(\theta)) \end{pmatrix}, \begin{pmatrix} \partial f_1/\partial \theta_j(\theta) \\ \vdots \\ \partial f_n/\partial \theta_j(\theta) \end{pmatrix} \right\rangle \neq 0$, $\sum_{i=1}^{N} w_i \Phi'(f_i(\theta)) \nabla f_i(\theta) = \nabla \left( \sum_{i=1}^{N} w_i \Phi(f_i(\theta)) \right) \neq 0$ for any $\theta \neq \theta^* \in \Omega$. Therefore, $\theta^*$ is the only critical point of $\sum_{i=1}^{N} w_i \Phi(f_i(\cdot))$ on $\Omega$. Since $\tilde{\theta}$ is a critical point on $\Omega$, we can conclude that $\theta^* = \tilde{\theta}$, and so $\theta^*$ is a minimizer of $\sum_{i=1}^{N} w_i \Phi(f_i(\cdot))$ on $\Omega$. Since any other minimizer must be a critical point and $\theta^*$ is the only critical point, $\theta^*$ is the unique minimizer. So, $\arg\min_{\theta \in \Omega} \sum_{i=1}^{N} f_i(\theta) = \theta^* = \arg\min_{\theta \in \Omega} \sum_{i=1}^{N} w_i \Phi(f_i(\theta))$. $\square$

**Lemma 2.** *Let $P$ be a distribution on $\mathbb{R}^d$ whose density $p(\cdot)$ is continuous at a point $\mathbf{x}_0 \in \mathbb{R}^d$ and $\mathbf{x} \sim P$ be a random variable. Let $\tilde{r} := \|\mathbf{x} - \mathbf{x}_0\|_2$, $\kappa := \pi^{d/2}/\Gamma\left(\frac{d}{2} + 1\right)$, where $\Gamma(\cdot)$ denotes the gamma function [1], and $r := \kappa \tilde{r}^d$. Let $G(\cdot)$ denote the cumulative distribution function (CDF) of $r$ and $\partial_+ G(\cdot)$ denote the one-sided derivative of $G$ from the right. Then, $\partial_+ G(0) = p(\mathbf{x}_0)$.*

*Proof.* By definition of $\partial_+ G(\cdot)$,

$$\partial_+ G(0) = \lim_{h \to 0^+} \frac{G(h) - G(0)}{h} = \lim_{h \to 0^+} \frac{G(h)}{h}$$
$$= \lim_{h \to 0^+} \frac{\Pr(r \leq h)}{h} = \lim_{h \to 0^+} \frac{\Pr\left(\tilde{r} \leq \sqrt[d]{h/\kappa}\right)}{h}$$

---

[1] The constant $\kappa$ is the the ratio of the volume of a $d$-dimensional ball of radius $\tilde{r}$ to a $d$-dimensional cube of side length $\tilde{r}$.

If we define $\tilde{h} := \sqrt[d]{h/\kappa}$, the above can be re-written as:

$$\partial_+ G(0) = \lim_{\tilde{h} \to 0^+} \frac{\Pr\left(\tilde{r} \leq \tilde{h}\right)}{\kappa \tilde{h}^d} = \lim_{\tilde{h} \to 0^+} \frac{\int_{B_{\mathbf{x}_0}(\tilde{h})} p(\mathbf{u}) d\mathbf{u}}{\kappa \tilde{h}^d}$$

We want to show that $\lim_{\tilde{h} \to 0^+} \left(\int_{B_{\mathbf{x}_0}(\tilde{h})} p(\mathbf{u}) d\mathbf{u}\right) / \kappa \tilde{h}^d = p(\mathbf{x}_0)$. In other words, we want to show $\forall \epsilon > 0 \; \exists \delta > 0$ such that $\forall \tilde{h} \in (0, \delta), \left| \frac{\int_{B_{\mathbf{x}_0}(\tilde{h})} p(\mathbf{u}) d\mathbf{u}}{\kappa \tilde{h}^d} - p(\mathbf{x}_0) \right| < \epsilon$.

Let $\epsilon > 0$ be arbitrary.

Since $p(\cdot)$ is continuous at $\mathbf{x}_0$, by definition, $\forall \tilde{\epsilon} > 0 \; \exists \tilde{\delta} > 0$ such that $\forall \mathbf{u} \in B_{\mathbf{x}_0}(\tilde{\delta})$, $|p(\mathbf{u}) - p(\mathbf{x}_0)| < \tilde{\epsilon}$. Let $\tilde{\delta} > 0$ be such that $\forall \mathbf{u} \in B_{\mathbf{x}_0}(\tilde{\delta})$, $p(\mathbf{x}_0) - \epsilon < p(\mathbf{u}) < p(\mathbf{x}_0) + \epsilon$. We choose $\delta = \tilde{\delta}$.

Let $0 < \tilde{h} < \delta$ be arbitrary. Since $p(\mathbf{x}_0) - \epsilon < p(\mathbf{u}) < p(\mathbf{x}_0) + \epsilon \; \forall \mathbf{u} \in B_{\mathbf{x}_0}(\tilde{\delta}) = B_{\mathbf{x}_0}(\delta) \supset B_{\mathbf{x}_0}(\tilde{h})$,

$$\int_{B_{\mathbf{x}_0}(\tilde{h})} p(\mathbf{u}) d\mathbf{u} < \int_{B_{\mathbf{x}_0}(\tilde{h})} \left(p(\mathbf{x}_0) + \epsilon\right) d\mathbf{u}$$
$$= \left(p(\mathbf{x}_0) + \epsilon\right) \int_{B_{\mathbf{x}_0}(\tilde{h})} d\mathbf{u}$$

Observe that $\int_{B_{\mathbf{x}_0}(\tilde{h})} d\mathbf{u}$ is the volume of a $d$-dimensional ball of radius $\tilde{h}$, so $\int_{B_{\mathbf{x}_0}(\tilde{h})} d\mathbf{u} = \kappa \tilde{h}^d$. Thus, $\int_{B_{\mathbf{x}_0}(\tilde{h})} p(\mathbf{u}) d\mathbf{u} < \kappa \tilde{h}^d \left(p(\mathbf{x}_0) + \epsilon\right)$, implying that $\left(\int_{B_{\mathbf{x}_0}(\tilde{h})} p(\mathbf{u}) d\mathbf{u}\right) / \kappa \tilde{h}^d < p(\mathbf{x}_0) + \epsilon$. By similar reasoning, we conclude that $\left(\int_{B_{\mathbf{x}_0}(\tilde{h})} p(\mathbf{u}) d\mathbf{u}\right) / \kappa \tilde{h}^d > p(\mathbf{x}_0) - \epsilon$.

Hence,

$$\left| \frac{\int_{B_{\mathbf{x}_0}(\tilde{h})} p(\mathbf{u}) d\mathbf{u}}{\kappa \tilde{h}^d} - p(\mathbf{x}_0) \right| < \epsilon \; \forall \tilde{h} \in (0, \delta)$$

Therefore,

$$\partial_+ G(0) = \lim_{\tilde{h} \to 0^+} \frac{\int_{B_{\mathbf{x}_0}(\tilde{h})} p(\mathbf{u}) d\mathbf{u}}{\kappa \tilde{h}^d} = p(\mathbf{x}_0)$$

$\square$

**Lemma 3.** *Let $P_\theta$ be a parameterized family of distributions on $\mathbb{R}^d$ with parameter $\theta$ and probability density function (PDF) $p_\theta(\cdot)$ that is continuous at a point $\mathbf{x}_i$. Consider a random variable $\tilde{\mathbf{x}}_1^\theta \sim P_\theta$ and define $\tilde{r}_i^\theta := \left\|\tilde{\mathbf{x}}_1^\theta - \mathbf{x}_i\right\|_2^2$, whose cumulative distribution function (CDF) is denoted by $F_i^\theta(\cdot)$. Assume $P_\theta$ has the following property: for any $\theta_1, \theta_2$, there exists $\theta_0$ such that $F_i^{\theta_0}(t) \geq \max\left\{F_i^{\theta_1}(t), F_i^{\theta_2}(t)\right\} \; \forall t \geq 0$ and $p_{\theta_0}(\mathbf{x}_i) = \max\left\{p_{\theta_1}(\mathbf{x}_i), p_{\theta_2}(\mathbf{x}_i)\right\}$. For any $m \geq 1$, let $\tilde{\mathbf{x}}_1^\theta, \ldots, \tilde{\mathbf{x}}_m^\theta \sim P_\theta$ be i.i.d. random variables and define $R_i^\theta := \min_{j \in [m]} \left\|\tilde{\mathbf{x}}_j^\theta - \mathbf{x}_i\right\|_2^2$. Then the function $\Psi_i : z \mapsto \min_\theta \left\{\mathbb{E}\left[R_i^\theta\right] | p_\theta(\mathbf{x}_i) = z\right\}$ is strictly decreasing.*

*Proof.* Let $r_i^\theta := \kappa \left(\tilde{r}_i^\theta\right)^{d/2} = \kappa \left\|\tilde{\mathbf{x}}_1^\theta - \mathbf{x}_i\right\|_2^d$ be a random variable and let $G_i^\theta(\cdot)$ be the CDF of $r_i^\theta$. Since $R_i^\theta$ is nonnegative,

$$
\begin{aligned}
\mathbb{E}\left[R_i^\theta\right] &= \int_0^\infty \Pr\left(R_i^\theta > t\right) dt \\
&= \int_0^\infty \left(\Pr\left(\left\|\tilde{\mathbf{x}}_1^\theta - \mathbf{x}_i\right\|_2^2 > t\right)\right)^m dt \\
&= \int_0^\infty \left(\Pr\left(\kappa \left\|\tilde{\mathbf{x}}_1^\theta - \mathbf{x}_i\right\|_2^d > \kappa t^{d/2}\right)\right)^m dt \\
&= \int_0^\infty \left(\Pr\left(r_i^\theta > \kappa t^{d/2}\right)\right)^m dt \\
&= \int_0^\infty \left(1 - G_i^\theta\left(\kappa t^{d/2}\right)\right)^m dt
\end{aligned}
$$

Also, by Lemma 2, $p_\theta(\mathbf{x}_i) = \partial_+ G_i^\theta(0)$. Using these facts, we can rewrite $\min_\theta \left\{\mathbb{E}\left[R_i^\theta\right] | p_\theta(\mathbf{x}_i) = z\right\}$ as $\min_\theta \left\{\int_0^\infty \left(1 - G_i^\theta\left(\kappa t^{d/2}\right)\right)^m dt \,|\, \partial_+ G_i^\theta(0) = z\right\}$. By definition of $\Psi_i$, $\min_\theta \left\{\int_0^\infty \left(1 - G_i^\theta\left(\kappa t^{d/2}\right)\right)^m dt \,|\, \partial_+ G_i^\theta(0) = z\right\}$ exists for all $z$. Let $\phi_i(z)$ be a value of $\theta$ that attains the minimum. Define $G_i^*(y, z) := G_i^{\phi_i(z)}(y)$. By definition, $\frac{\partial_+}{\partial y} G_i^*(0, z) = z$, where $\frac{\partial_+}{\partial y} G_i^*(y, z)$ denotes the one-sided partial derivative from the right w.r.t. $y$. Also, since $G_i^*(\cdot, z)$ is the CDF of a distribution of a non-negative random variable, $G_i^*(0, z) = 0$.

By definition of $\frac{\partial_+}{\partial y} G_i^*(0, z)$, $\forall \epsilon > 0 \,\exists \delta > 0$ such that $\forall h \in (0, \delta)$, $\left|\frac{G_i^*(h,z) - G_i^*(0,z)}{h} - z\right| < \epsilon$.

Let $z' > z$. Let $\delta > 0$ be such that $\forall h \in (0, \delta)$, $\left|\frac{G_i^*(h,z) - G_i^*(0,z)}{h} - z\right| < \frac{z'-z}{2}$ and $\delta' > 0$ be such that $\forall h \in (0, \delta')$, $\left|\frac{G_i^*(h,z') - G_i^*(0,z')}{h} - z'\right| < \frac{z'-z}{2}$.

Consider $h \in (0, \min(\delta, \delta'))$. Then, $\frac{G_i^*(h,z) - G_i^*(0,z)}{h} = \frac{G_i^*(h,z)}{h} < z + \frac{z'-z}{2} = \frac{z+z'}{2}$ and $\frac{G_i^*(h,z') - G_i^*(0,z')}{h} = \frac{G_i^*(h,z')}{h} > z' - \frac{z'-z}{2} = \frac{z+z'}{2}$. So,

$$
\frac{G_i^*(h, z)}{h} < \frac{z + z'}{2} < \frac{G_i^*(h, z')}{h}
$$

Multiplying by $h$ on both sides, we conclude that $G_i^*(h, z) < G_i^*(h, z') \,\forall h \in (0, \min(\delta, \delta'))$.

Let $\alpha := \sqrt[d]{\min(\delta, \delta')/\kappa}$. We can break $\int_0^\infty \left(1 - G_i^*\left(\kappa t^{d/2}, z\right)\right)^m dt$ into two terms:

$$
\begin{aligned}
&\int_0^\infty \left(1 - G_i^*\left(\kappa t^{d/2}, z\right)\right)^m dt \\
&= \int_0^\alpha \left(1 - G_i^*\left(\kappa t^{d/2}, z\right)\right)^m dt + \int_\alpha^\infty \left(1 - G_i^*\left(\kappa t^{d/2}, z\right)\right)^m dt
\end{aligned}
$$

We can also do the same for $\int_0^\infty \left(1 - G_i^*\left(\kappa t^{d/2}, z'\right)\right)^m dt$.

Because $G_i^*(h, z) < G_i^*(h, z') \,\forall h \in (0, \min(\delta, \delta'))$, $G_i^*(\kappa t^{d/2}, z) < G_i^*(\kappa t^{d/2}, z') \,\forall t \in (0, \alpha)$. It follows that $1 - G_i^*(\kappa t^{d/2}, z) > 1 - G_i^*(\kappa t^{d/2}, z')$ and $\left(1 - G_i^*(\kappa t^{d/2}, z)\right)^m > \left(1 - G_i^*(\kappa t^{d/2}, z')\right)^m \,\forall t \in (0, \alpha)$. So, $\int_0^\alpha \left(1 - G_i^*\left(\kappa t^{d/2}, z\right)\right)^m dt > \int_0^\alpha \left(1 - G_i^*\left(\kappa t^{d/2}, z'\right)\right)^m dt$.

We now consider the second term. First, observe that $F_i^\theta(t) = \Pr\left(\left\|\tilde{\mathbf{x}}_1^\theta - \mathbf{x}_i\right\|_2^2 \le t\right) = \Pr\left(\kappa \left\|\tilde{\mathbf{x}}_1^\theta - \mathbf{x}_i\right\|_2^d \le \kappa t^{d/2}\right) = G_i^\theta\left(\kappa t^{d/2}\right)$ for all $t \ge 0$. So, by the property of $P_\theta$, for any $\theta_1, \theta_2$, there exists $\theta_0$ such that $G_i^{\theta_0}(\kappa t^{d/2}) = F_i^{\theta_0}(t) \ge \max\left\{F_i^{\theta_1}(t), F_i^{\theta_2}(t)\right\} = \max\left\{G_i^{\theta_1}(\kappa t^{d/2}), G_i^{\theta_2}(\kappa t^{d/2})\right\} \,\forall t \ge 0$ and $\partial_+ G_i^{\theta_0}(0) = p_{\theta_0}(\mathbf{x}_i) = \max\left\{p_{\theta_1}(\mathbf{x}_i), p_{\theta_2}(\mathbf{x}_i)\right\} = \max\left\{\partial_+ G_i^{\theta_1}(0), \partial_+ G_i^{\theta_2}(0)\right\}$.

Take $\theta_1 = \phi_i(z)$ and $\theta_2 = \phi_i(z')$. Let $\theta_0$ be such that $G_i^{\theta_0}(\kappa t^{d/2}) \geq \max\left\{G_i^{\theta_1}(\kappa t^{d/2}), G_i^{\theta_2}(\kappa t^{d/2})\right\} \ \forall t \geq 0$ and $\partial_+ G_i^{\theta_0}(0) = \max\left\{\partial_+ G_i^{\theta_1}(0), \partial_+ G_i^{\theta_2}(0)\right\}$. By definition of $\phi_i(\cdot)$, $\partial_+ G_i^{\theta_1}(0) = z$ and $\partial_+ G_i^{\theta_2}(0) = z'$. So, $\partial_+ G_i^{\theta_0}(0) = \max\{z, z'\} = z'$. Since $G_i^{\theta_0}(\kappa t^{d/2}) \geq G_i^{\theta_2}(\kappa t^{d/2}) \ \forall t \geq 0$, $1 - G_i^{\theta_0}\left(\kappa t^{d/2}\right) \leq 1 - G_i^{\theta_2}\left(\kappa t^{d/2}\right) \ \forall t \geq 0$ and so $\int_0^\infty \left(1 - G_i^{\theta_0}\left(\kappa t^{d/2}\right)\right)^m dt \leq \int_0^\infty \left(1 - G_i^{\theta_2}\left(\kappa t^{d/2}\right)\right)^m dt$. On the other hand, because $\theta_2 = \phi_i(z')$ minimizes $\int_0^\infty \left(1 - G_i^{\theta}\left(\kappa t^{d/2}\right)\right)^m dt$ among all $\theta$'s such that $\partial_+ G_i^{\theta}(0) = z'$ and $\partial_+ G_i^{\theta_0}(0) = z'$, $\int_0^\infty \left(1 - G_i^{\theta_2}\left(\kappa t^{d/2}\right)\right)^m dt \leq \int_0^\infty \left(1 - G_i^{\theta_0}\left(\kappa t^{d/2}\right)\right)^m dt$. We can therefore conclude that $\int_0^\infty \left(1 - G_i^{\theta_0}\left(\kappa t^{d/2}\right)\right)^m dt = \int_0^\infty \left(1 - G_i^{\theta_2}\left(\kappa t^{d/2}\right)\right)^m dt$. Since $1 - G_i^{\theta_0}\left(\kappa t^{d/2}\right) \leq 1 - G_i^{\theta_2}\left(\kappa t^{d/2}\right) \ \forall t \geq 0$, the only situation where this can happen is when $G_i^{\theta_0}\left(\kappa t^{d/2}\right) = G_i^{\theta_2}\left(\kappa t^{d/2}\right) \ \forall t \geq 0$.

By definition of $G_i^*$, $G_i^*\left(\kappa t^{d/2}, z\right) = G_i^{\phi_i(z)}(\kappa t^{d/2}) = G_i^{\theta_1}(\kappa t^{d/2})$ and $G_i^*\left(\kappa t^{d/2}, z'\right) = G_i^{\phi_i(z')}(\kappa t^{d/2}) = G_i^{\theta_2}(\kappa t^{d/2}) = G_i^{\theta_0}\left(\kappa t^{d/2}\right)$. By definition of $\theta_0$, $G_i^{\theta_0}\left(\kappa t^{d/2}\right) \geq G_i^{\theta_1}(\kappa t^{d/2}) \ \forall t \geq 0$. So, $G_i^*\left(\kappa t^{d/2}, z'\right) = G_i^{\theta_2}(\kappa t^{d/2}) \geq G_i^{\theta_1}(\kappa t^{d/2}) = G_i^*\left(\kappa t^{d/2}, z\right) \ \forall t \geq 0$. Hence, $\int_\alpha^\infty \left(1 - G_i^*\left(\kappa t^{d/2}, z'\right)\right)^m dt \leq \int_\alpha^\infty \left(1 - G_i^*\left(\kappa t^{d/2}, z\right)\right)^m dt$.

Combining with the previous result that $\int_0^\alpha \left(1 - G_i^*\left(\kappa t^{d/2}, z'\right)\right)^m dt < \int_0^\alpha \left(1 - G_i^*\left(\kappa t^{d/2}, z\right)\right)^m dt$, it follows that:

$$\int_0^\infty \left(1 - G_i^*\left(\kappa t^{d/2}, z'\right)\right)^m dt$$
$$= \int_0^\alpha \left(1 - G_i^*\left(\kappa t^{d/2}, z'\right)\right)^m dt + \int_\alpha^\infty \left(1 - G_i^*\left(\kappa t^{d/2}, z'\right)\right)^m dt$$
$$< \int_0^\alpha \left(1 - G_i^*\left(\kappa t^{d/2}, z\right)\right)^m dt + \int_\alpha^\infty \left(1 - G_i^*\left(\kappa t^{d/2}, z'\right)\right)^m dt$$
$$\leq \int_0^\alpha \left(1 - G_i^*\left(\kappa t^{d/2}, z\right)\right)^m dt + \int_\alpha^\infty \left(1 - G_i^*\left(\kappa t^{d/2}, z\right)\right)^m dt$$
$$= \int_0^\infty \left(1 - G_i^*\left(\kappa t^{d/2}, z\right)\right)^m dt$$

By definition,

$$\int_0^\infty \left(1 - G_i^*\left(\kappa t^{d/2}, z\right)\right)^m dt$$
$$= \int_0^\infty \left(1 - G_i^{\phi_i(z)}(\kappa t^{d/2})\right)^m dt$$
$$= \min_\theta \left\{\int_0^\infty \left(1 - G_i^{\theta}\left(\kappa t^{d/2}\right)\right)^m dt \,\Big|\, \partial_+ G_i^{\theta}(0) = z\right\}$$
$$= \min_\theta \left\{\mathbb{E}\left[R_i^{\theta}\right] \,|\, p_\theta(\mathbf{x}_i) = z\right\}$$
$$= \Psi_i(z)$$

Similarly, $\int_0^\infty \left(1 - G_i^*\left(\kappa t^{d/2}, z'\right)\right)^m dt = \Psi_i(z')$. We can therefore conclude that $\Psi_i(z') < \Psi_i(z)$ whenever $z' > z$. $\qquad \square$

We now prove the main result.

**Theorem 1.** *Consider a set of observations $\mathbf{x}_1, \ldots, \mathbf{x}_n$, a parameterized family of distributions $P_\theta$ with probability density function (PDF) $p_\theta(\cdot)$ and a unique maximum likelihood solution $\theta^*$. For any $m \geq 1$, let $\tilde{\mathbf{x}}_1^\theta, \ldots, \tilde{\mathbf{x}}_m^\theta \sim P_\theta$ be i.i.d. random variables and define $\tilde{r}^\theta := \left\|\tilde{\mathbf{x}}_1^\theta\right\|_2^2$, $R^\theta := \min_{j \in [m]} \left\|\tilde{\mathbf{x}}_j^\theta\right\|_2^2$ and $R_i^\theta := \min_{j \in [m]} \left\|\tilde{\mathbf{x}}_j^\theta - \mathbf{x}_i\right\|_2^2$. Let $F^\theta(\cdot)$ be the cumulative distribution function (CDF) of $\tilde{r}^\theta$ and $\Psi(z) := \min_\theta \left\{\mathbb{E}\left[R^\theta\right] \,|\, p_\theta(\mathbf{0}) = z\right\}$.*

*If $P_\theta$ satisfies the following:*

- $p_\theta(\mathbf{x})$ *is differentiable w.r.t.* $\theta$ *and continuous w.r.t.* $\mathbf{x}$ *everywhere.*

- $\forall \theta, \mathbf{v}$, *there exists* $\theta'$ *such that* $p_\theta(\mathbf{x}) = p_{\theta'}(\mathbf{x} + \mathbf{v})$ $\forall \mathbf{x}$.

- *For any* $\theta_1, \theta_2$, *there exists* $\theta_0$ *such that* $F^{\theta_0}(t) \geq \max\left\{F^{\theta_1}(t), F^{\theta_2}(t)\right\}$ $\forall t \geq 0$ *and* $p_{\theta_0}(\mathbf{0}) = \max\left\{p_{\theta_1}(\mathbf{0}), p_{\theta_2}(\mathbf{0})\right\}$.

- $\exists \tau > 0$ *such that* $\forall i \in [n]$ $\forall \theta \notin B_{\theta^*}(\tau)$, $p_\theta(\mathbf{x}_i) < p_{\theta*}(\mathbf{x}_i)$, *where* $B_{\theta^*}(\tau)$ *denotes the ball centred at* $\theta^*$ *of radius* $\tau$.

- $\Psi(z)$ *is differentiable everywhere.*

- *For all* $\theta$, *if* $\theta \neq \theta^*$, *there exists* $j \in [d]$ *such that*
$$\left\langle \begin{pmatrix} \frac{\Psi'(p_\theta(\mathbf{x}_1))p_\theta(\mathbf{x}_1)}{\Psi'(p_{\theta^*}(\mathbf{x}_1))p_{\theta^*}(\mathbf{x}_1)} \\ \vdots \\ \frac{\Psi'(p_\theta(\mathbf{x}_n))p_\theta(\mathbf{x}_n)}{\Psi'(p_{\theta^*}(\mathbf{x}_n))p_{\theta^*}(\mathbf{x}_n)} \end{pmatrix}, \begin{pmatrix} \nabla_\theta\left(\log p_\theta(\mathbf{x}_1)\right)_j \\ \vdots \\ \nabla_\theta\left(\log p_\theta(\mathbf{x}_n)\right)_j \end{pmatrix} \right\rangle \neq 0.$$

*Then,*
$$\arg\min_\theta \sum_{i=1}^n \frac{\mathbb{E}\left[R_i^\theta\right]}{\Psi'(p_{\theta^*}(\mathbf{x}_i))p_{\theta^*}(\mathbf{x}_i)} = \arg\max_\theta \sum_{i=1}^n \log p_\theta(\mathbf{x}_i)$$

*Furthermore, if* $p_{\theta^*}(\mathbf{x}_1) = \cdots = p_{\theta^*}(\mathbf{x}_n)$, *then,*
$$\arg\min_\theta \sum_{i=1}^n \mathbb{E}\left[R_i^\theta\right] = \arg\max_\theta \sum_{i=1}^n \log p_\theta(\mathbf{x}_i)$$

*Proof.* Pick an arbitrary $i \in [n]$. We first prove a few basic facts.

By the second property of $P_\theta$, $\forall \theta$ $\exists \theta'$ such that $p_\theta(\mathbf{u}) = p_{\theta'}(\mathbf{u} - \mathbf{x}_i)$ $\forall \mathbf{u}$. In particular, $p_\theta(\mathbf{x}_i) = p_{\theta'}(\mathbf{x}_i - \mathbf{x}_i) = p_{\theta'}(\mathbf{0})$. Let $F_i^\theta$ be as defined in Lemma 3.

$$F_i^\theta(t) = \Pr\left(\tilde{r}_i^\theta \leq t\right) = \Pr\left(\left\|\tilde{\mathbf{x}}_1^\theta - \mathbf{x}_i\right\|_2 \leq \sqrt{t}\right)$$
$$= \int_{B_{\mathbf{x}_i}(\sqrt{t})} p_\theta(\mathbf{u})d\mathbf{u} = \int_{B_{\mathbf{x}_i}(\sqrt{t})} p_{\theta'}(\mathbf{u} - \mathbf{x}_i)d\mathbf{u}$$
$$= \int_{B_{\mathbf{0}}(\sqrt{t})} p_{\theta'}(\mathbf{u})d\mathbf{u} = \Pr\left(\tilde{r}^{\theta'} \leq t\right) = F^{\theta'}(t)$$

Similarly, $\forall \theta'$ $\exists \theta$ such that $p_{\theta'}(\mathbf{u}) = p_\theta(\mathbf{u} + \mathbf{x}_i)$ $\forall \mathbf{u}$. In particular, $p_{\theta'}(\mathbf{0}) = p_\theta(\mathbf{0} + \mathbf{x}_i) = p_\theta(\mathbf{x}_i)$.

$$F^{\theta'}(t) = \Pr\left(\tilde{r}^{\theta'} \leq t\right) = \int_{B_{\mathbf{0}}(\sqrt{t})} p_{\theta'}(\mathbf{u})d\mathbf{u}$$
$$= \int_{B_{\mathbf{0}}(\sqrt{t})} p_\theta(\mathbf{u} + \mathbf{x}_i)d\mathbf{u} = \int_{B_{\mathbf{x}_i}(\sqrt{t})} p_\theta(\mathbf{u})d\mathbf{u}$$
$$= \Pr\left(\left\|\tilde{\mathbf{x}}_1^\theta - \mathbf{x}_i\right\|_2 \leq \sqrt{t}\right) = \Pr\left(\tilde{r}_i^\theta \leq t\right) = F_i^\theta(t)$$

Let $\theta_1, \theta_2$ be arbitrary. The facts above imply that there exist $\theta_1'$ and $\theta_2'$ such that $F_i^{\theta_1}(t) = F^{\theta_1'}(t)$, $F_i^{\theta_2}(t) = F^{\theta_2'}(t)$, $p_{\theta_1}(\mathbf{x}_i) = p_{\theta_1'}(\mathbf{0})$ and $p_{\theta_2}(\mathbf{x}_i) = p_{\theta_2'}(\mathbf{0})$.

By the third property of $P_\theta$, let $\theta_0'$ be such that $F^{\theta_0'}(t) \geq \max\left\{F^{\theta_1'}(t), F^{\theta_2'}(t)\right\}$ $\forall t \geq 0$ and $p_{\theta_0'}(\mathbf{0}) = \max\left\{p_{\theta_1'}(\mathbf{0}), p_{\theta_2'}(\mathbf{0})\right\}$. By the facts above, it follows that there exists $\theta_0$ such that $F^{\theta_0'}(t) = F_i^{\theta_0}(t)$ and $p_{\theta_0'}(\mathbf{0}) = p_{\theta_0}(\mathbf{x}_i)$.

So, we can conclude that for any $\theta_1, \theta_2$, there exists $\theta_0$ such that $F_i^{\theta_0}(t) \geq \max\left\{F_i^{\theta_1}(t), F_i^{\theta_2}(t)\right\}$ $\forall t \geq 0$ and $p_{\theta_0}(\mathbf{x}_i) = \max\left\{p_{\theta_1}(\mathbf{x}_i), p_{\theta_2}(\mathbf{x}_i)\right\}$.

By Lemma 3, $\Psi_i(z) = \min_\theta \left\{ \mathbb{E}\left[R_i^\theta\right] | p_\theta(\mathbf{x}_i) = z \right\}$ is strictly decreasing.

Consider any $\theta$. By the facts above, there exists $\theta'$ such that $p_\theta(\mathbf{x}_i) = p_{\theta'}(\mathbf{0})$ and $F_i^\theta(t) = F^{\theta'}(t)$ $\forall t$. Therefore,

$$
\begin{aligned}
\mathbb{E}\left[R_i^\theta\right] &= \int_0^\infty \Pr\left(R_i^\theta > t\right) dt \\
&= \int_0^\infty \left(\Pr\left(\left\|\tilde{\mathbf{x}}_1^\theta - \mathbf{x}_i\right\|_2^2 > t\right)\right)^m dt \\
&= \int_0^\infty \left(1 - F_i^\theta(t)\right)^m dt \\
&= \int_0^\infty \left(1 - F^{\theta'}(t)\right)^m dt \\
&= \int_0^\infty \Pr\left(R^{\theta'} > t\right) dt \\
&= \mathbb{E}\left[R^{\theta'}\right]
\end{aligned}
$$

So, $\forall z$

$$
\begin{aligned}
\Psi_i(z) &= \min_\theta \left\{ \mathbb{E}\left[R_i^\theta\right] | p_\theta(\mathbf{x}_i) = z \right\} \\
&= \min_{\theta'} \left\{ \mathbb{E}\left[R^{\theta'}\right] | p_{\theta'}(\mathbf{0}) = z \right\} \\
&= \Psi(z)
\end{aligned}
$$

Because $\Psi_i(\cdot)$ is strictly decreasing, $\Psi(\cdot)$ is also strictly decreasing.

We would like to apply Lemma 1, with $f_i(\theta) = -\log p_\theta(\mathbf{x}_i)$ $\forall i \in [n]$ and $\Phi(y) = \Psi(\exp(-y))$. By the first property of $P_\theta$, $p_\theta(\cdot)$ is differentiable w.r.t. $\theta$ and so $f_i(\theta)$ is differentiable for all $i$. By the fifth property of $P_\theta$, $\Psi(\cdot)$ is differentiable and so $\Phi(\cdot)$ is differentiable. Since $y \mapsto \exp(-y)$ is strictly decreasing and $\Psi(\cdot)$ is strictly decreasing, $\Phi(\cdot)$ is strictly increasing. Since there is a unique maximum likelihood solution $\theta^*$, $\min_\theta \sum_{i=1}^n f_i(\theta) = \max_\theta \sum_{i=1}^n \log p_\theta(\mathbf{x}_i)$ exists and has a unique minimizer. By the fourth property of $P_\theta$, the first condition of Lemma 1 is satisfied. By the sixth property of $P_\theta$, the second condition of Lemma 1 is satisfied. Since all conditions are satisfied, we apply Lemma 1 and conclude that

$$
\begin{aligned}
\min_\theta \sum_{i=1}^n w_i \Phi(f_i(\theta)) &= \min_\theta \sum_{i=1}^n w_i \Psi(p_\theta(\mathbf{x}_i)) \\
&= \min_\theta \sum_{i=1}^n w_i \Psi_i(p_\theta(\mathbf{x}_i)) \\
&= \min_\theta \sum_{i=1}^n \frac{\mathbb{E}\left[R_i^\theta\right]}{\Psi'(p_{\theta^*}(\mathbf{x}_i))p_{\theta^*}(\mathbf{x}_i)}
\end{aligned}
$$

exists and has a unique minimizer. Furthermore,

$$
\begin{aligned}
\arg\min_\theta \sum_{i=1}^n \frac{\mathbb{E}\left[R_i^\theta\right]}{\Psi'(p_{\theta^*}(\mathbf{x}_i))p_{\theta^*}(\mathbf{x}_i)} &= \arg\min_\theta \sum_{i=1}^n -\log p_\theta(\mathbf{x}_i) \\
&= \arg\max_\theta \sum_{i=1}^n \log p_\theta(\mathbf{x}_i)
\end{aligned}
$$

If $p_\theta(\mathbf{x}_1) = \cdots p_\theta(\mathbf{x}_n)$, then $w_1 = \cdots = w_n$, and so $\arg\min_\theta \sum_{i=1}^n w_i \mathbb{E}\left[R_i^\theta\right] = \arg\min_\theta \sum_{i=1}^n \mathbb{E}\left[R_i^\theta\right] = \arg\max_\theta \sum_{i=1}^n \log p_\theta(\mathbf{x}_i)$. $\qquad\square$

