# OpenReview forum: "Implicit Maximum Likelihood Estimation"
_ICLR.cc/2019/Conference_

### Official Review · AnonReviewer2 · 2018-11-01
**Nice Theory, Questionable Practicality**

**Rating:** 5
**Confidence:** 4

**Review:**

Summary:

This paper proposes a nearest-neighbor-based algorithm for implicit maximum likelihood.  Samples are produced by the generator network and then a nearest neighbors algorithm is run to match the samples with their nearest data point.  The generator is then updated using the Euclidean distance between samples and neighbors as the optimization objective.  Six conditions are then provided, and if they are met, then the authors show that this method is performing maximum likelihood on the implied density.  Experiments report Parzen window density estimates, samples from the model, and latent-space interpolations for MNIST, Toronto Faces, and CIFAR-10.

Pros:

The primary contribution of this paper is an algorithm for implicit likelihood maximization with theoretic guarantees.  As far as I’m aware, this is a novel and noteworthy contribution.  Moreover, as each sample must be paired with an observation, it does seem like the algorithm would be somewhat robust to the notorious mode collapse problem.

Cons:

My primary critique of the paper is that there is very little experimental investigation of the crucial details of the algorithm.  Firstly, running the nearest neighbors algorithm seems like it could be a computational bottleneck.  The authors acknowledge this, but then say “this is no longer the case due to recent advances in nearest neighbor search algorithms (Li & Malik 2016; 2017)” (p 3-4).  No other justification is given, from what I can tell.  A simulation showing how the runtime scales with dimensionality or number of data points would be very useful for knowing the scalability and practicality of the algorithm.  In the same vein, showing that the algorithm works well even with a relaxation such as approximate neighbors or random projections would make the algorithm more attractive to adopt.

Moreover, I found it frustrating that the paper teases a fix to several well-known GAN issues: “The proposed method could sidestep the three issues mentioned above: mode collapse, vanishing gradients and training instability” (p 3).  But the paper never experimentally investigates if the proposed approach indeed is better in these aspects.  I was disappointed since, intuitively, the algorithm does seem like it could be robust to mode collapse.  In addition to this lack of experimental focus, the only quantitative result is the Parzen window estimates in Table 1.  The proposed method does best the others but the other reported results are quite old---all from 2015 or earlier.

Minor points:

The paper is at 10 pages, and while it is well-written, the writing is verbose and could be use tightening.


Evaluation:  This paper presents an interesting contribution: an implicit likelihood estimation algorithm amenable to theoretical analysis.  Moreover, the theory seems not too divorced from practice (but I didn't check every detail).  However, the evaluation of this method is where the paper falters.  A big issue (that the authors note themselves) is the practicality of performing repeated nearest neighbor iterations.  No runtimes are report, nor are any approximations considered.  Rather, samples and interpolations are given the most discussion.  Furthermore, there is no demonstrations of training stability or quantitative analysis of mode collapse.  Due to these experimental deficiencies, I recommend rejection, weakly.

---

### Official Review · AnonReviewer3 · 2018-11-01
**Interesting idea, but there is room for improving the presentation and the strength of the results**

**Rating:** 3
**Confidence:** 4

**Review:**

The paper proposes a new algorithm for implicit maximum likelihood estimation based on a fast Nearest Neighbor search. The algorithm can be used to implicitly maximize the likelihood of models for which the former quantity is not intractable but for which sampling is easy which is typically the case for implicit models.  The paper shows that under some conditions the optimal solution of the algorithm corresponds to the MLE solution and provides some experimental evidence that the method leads to a higher likelihood. However, The paper lacks clarity and the experiments are not really convincing. Here are some remarks:
Experiments:
- The estimated likelihood was reported on table 1 using parson window which is known to have bad scaling behavior with the dimension of data. In the end, the table compares methods that maximize different objectives and are evaluated with an unreliable metric. Here are two possible experiments that could be more informative:
- Consider toy examples for which the likelihood can be evaluated and the MLE obtained easily and then compare with the proposed method. This would already give a good sense of how well the algorithm behaves in simple cases.
- Another possibility is to use generative models like Real-NVP for which the likelihood can also be computed in closed form. This would allow comparing the proposed algorithm to direct likelihood maximization on more complicated datasets as done in [1].
It seems like having experiments of this nature is far more convincing than a long justification for why the results are not necessarily state-of-the-art.
- There are way too many samples on the figures so it is very hard to perform any visual assessment.

Theory:
- More discussions of the assumptions are needed, concrete examples for which these assumptions hold or not would be very useful.
- Lemma 2 is a direct consequence of the following result: if p is continuous at x_0 then x_0 is a Lebesgue point.

General remarks on the paper:
- What complexity is the nearest neighbor algorithm? Since it is crucial for the proposed method to be scalable it is worth presenting this algorithm at a high level in the main paper.
- The discussion in section 3 could be much more concise if concrete examples and figures were provided. Most of the facts discussed in that section are generally well understood, so conciseness is very appreciated in this case.
- «  A secondary issue that is more easily solvable is that samples presented in papers are sometimes cherry-picked; as a result, they capture the maximum sample quality, but not necessarily the mean sample quality. » Could you please provide an example of such paper? I would be very interested in having a closer look.
- In the last paragraph of section 5, it is said that although the samples may not be state of the art in terms of precision, other methods which achieve better precision «  may » have less recall. It would be good to have empirical evidence to back this claim.


Revision: Although this paper presents an interesting idea, there is a serious lack of evidence to support the claims in the paper:
- Missing experimental evidence for the efficiency of the NN search algorithm.
- Experiments are using Parzen window for estimating likelihood which  are known to be unreliable in high dimensions.
- None of the suggested experiments were considered. In my opinion these experiments could improve the quality of this work.
- Moreover, as mentioned by reviewer 1, Grover et al., 2017 provides evidence contrary to what the authors claim but this was never addressed so far in the paper.
- Theorem 1 makes rather strong assumptions: as pointed out by reviewer 1, assumption 3 is unlikely to hold for the distributions used in practice

For these reasons I recommend a clear reject.

[1] I. Danihelka, B. Lakshminarayanan, B. Uria, D. Wierstra, and P. Dayan. Comparison of Maximum Likelihood and GAN-based training of Real NVPs.

---

### Official Review · AnonReviewer1 · 2018-11-02
**Novel and interesting idea, but significant algorithmic and empirical concerns**

**Rating:** 4
**Confidence:** 4

**Review:**

Two high-level points about my review before going into the details:
1. This paper was a thoroughly enjoyable and insightful read. Kudos to the authors for attempting such a comprehensive overview of likelihood-based vs. likelihood-free learning.
2. I’ll be more than happy to revise my current rating if my concerns are addressed by the authors.

With regards to the technical assessment of this work, the idea of using a nearest neighbors objective for learning a generative model is both intriguing and appealing. What makes this work even more interesting are its connections with maximum likelihood estimation. Novelty aside, I believe there are major theoretical, algorithmic, and empirical concerns in the current work which I discuss below:

Theorem 1
- The third condition is true for location-scale family of distributions e.g., Gaussian. But the distribution learned by a generative model p_theta is far from Gaussian or other location-scale distributions.
- More importantly, I don’t think the upper bound is tight in practice because the likelihoods can vary significantly across the dataset. Take MNIST for example. Compare the log-likelihoods of an autoregressive model or ELBOs of a VAE across the different classes of digits. Straight digits (like 1s) have much higher log-likelihoods on average than curved digits.

Algorithm
- While significant advancements have indeed been made for nearest neighbor evaluation as the authors highlight, it’s hard to believe without any empirical evidence that nearest neighbor evaluation is indeed efficient in comparison to other methods of likelihood evaluation.
- Similarly, I was a bit disappointed by the choice of Euclidean distance in a pixel space as the choice of distance metric. The argument that you do not want to use “auxiliary sources of labelled data or leverage domain-specific prior knowledge” is indeed necessary for fair comparisons, but also points to a limitation of the current approach.

Empirical evaluation
- Seems too outdated both in terms of baselines and metrics. The authors are clearly aware of the current research in generative modeling but the current work provides almost no strong evidence to consider this work as an alternative to other approaches.
- While it is arguably well-established that Parzen window estimates are misleading (Theis et al.), that’s the only quantitative estimate in this work (Table 1). Hard to think of any recent published work (last 1-2 years) in generative modeling that even reports these estimates.
- The baselines in Table 1 are all from 2013-15. Clearly, much has happened in the last 3 years that merit the inclusion of more recent baselines.
-  Even for sample quality, there has been a lot of research in designing and improving metrics. E.g., Inception scores, Frechet Inception Distance, Kernel Inception Distance. I am not looking for state-of-the-art numbers, showing heavily zoomed out samples without any of these metrics is slightly disingenuous.
- As mentioned before, reporting the computation time/per iteration and number of iterations for convergence for the proposed algorithm in comparison with other approaches  is important.
- Similarly, the argument about the method avoiding even the other GAN problems (e.g., vanishing gradients, stability in training) can and should be supported by empirical evidence.

Analysis and discussion
- One family of generative models that is crucially missing from this work is normalizing flow models.
- This is somewhat debatable, but I do not agree that the tradeoff between likelihoods and sample quality is due to model capacity. As far as I can tell, the cited work of Grover et al., 2017 provides evidence contrary to what the authors claim. The prior work trained the same normalizing flow model via maximum likelihood and adversarial training, and observed vastly different results on likelihood and sample quality metrics. So model capacity isn't necessarily the key differentiating factor (which is same for both training algorithms in their experiments), it's more about the choice of the objective function and the optimization procedure.

Minor points for improving presentation:
- Section 3 can be made more concise and to the point. I’d be especially interested if the precision and recall discussion in this section and elsewhere can be formalized.
- Use numbered lists instead of bullets for assumptions in Theorem 1, so that the discussion of the assumptions right after the theorem statement are easy to follow.
- The citation of Grover et al. seems outdated? The current title is Flow-GAN: Combining maximum likelihood and adversarial learning in generative models.
- In general, avoid making somewhat hard assertions that are speculative. Some of them I’ve highlighted earlier in my review (e.g., some of the theorem assumptions being typically true, comparison of likelihood and sample quality based on model capacity etc.).

---

### Public Comment · (anonymous) · 2018-10-23
**The method is almost identical to "Generative Latent Optimization"**

As far as I understand, the only difference between the proposed method and the Generative Latent Optimization(https://arxiv.org/abs/1707.05776 , published at ICML 2018) is a very minor detail. The GLO optimizes for the latent codes whereas the submission keeps them fixed. The rest of the method is exactly same. Am I missing something?

If I am correct; the authors should discuss the differences properly, cite the paper and provide an empirical comparison.

---

> ### Author Response · Authors · 2018-10-25
> **It's actually quite different**
>
> That's actually not correct. There are two major differences:
>
> (1) GLO is not a probabilistic model, because there is no probability density associated with the latent vectors z. Instead it simply learns a mapping from some latent space to the output space, in the hope that the latent space is easier to model than the original output space (in this sense, it is more closely related to dimensionality reduction methods like PCA and autoencoders). In order to generate novel examples, one still needs to fit a probabilistic model to the learned latent vectors. In the case of GLO, a Gaussian is used for this purpose. In contrast, the proposed method trains a stand-alone probabilistic model, and so there is no need to fit another probabilistic model.
>
> (2) GLO enforces a 1-1 mapping between the latent vectors z and the data examples, whereas the proposed method does not. Not enforcing a 1-1 mapping is critical for showing equivalence to maximum likelihood, because of the asymmetric nature of KL-divergence. More specifically, maximum likelihood corresponds to minimizing D_KL(data || model), and it is well-known that minimizing D_KL(model || data), which swaps the data and the model distributions, is *not* equivalent to maximum likelihood. Now, suppose that we enforce a 1-1 mapping between the latent vectors and the data examples, then swapping the latent vectors and data examples in the proposed loss function would not change the loss function. This shows that if we were to enforce a 1-1 mapping, minimizing the loss cannot be equivalent to maximum likelihood.

---

### Author Response · Authors · 2018-12-03
**Rebuttal**

Sorry for the delay in posting the rebuttal. We've been a bit short on time due to various other deadlines, but below is a rebuttal of the key points that were raised.

AnonReviewer1:

Theorem1: Any distribution that can be translated and scaled arbitrarily is in a location-sacle family of distributions. This is certainly true of neural nets applied to noise from a fixed distribution, since the biases can be adjusted to translate arbitrarily, and the weights can be adjusted to scale arbitrarily.
Different models can assign very different log-likelihoods to the same data, and what log-likelihood an autoregressive model or a VAE assigns has no bearing on what a different model assigns at the maximum likelihood estimate. (Moreover, a VAE does not necessarily find the maximum likelihood estimate because the variational family may not contain the true posterior.)

Performance of the particular nearest neighbour search method was reported in the paper that describes it. The code is also publicly available, so you may also test it yourself. In the context of our method, we performed nearest neighbour search for 8,000 queries over 200,000 samples, each of which is 3072-dimensional. Constructing the data structure took 8.01 seconds, and querying took 1.31 seconds on a 4-year-old six-core CPU. This is relatively insignificant compared to the amount of time taken by backpropagation, which takes 181.85 seconds for 100 iterations of SGD on a 1080 Ti GPU.
We pointed out the fact that Euclidean distance can be applied to feature space in the last paragraph of section 2.2, so using Euclidean distance on pixel space does not point to a limitation. In fact, subsequent work on IMLE does this with ease, but that does not mean we should do the same in the original paper. In general, this is how science works - the initial paper on any given method should be applied to the most basic and generic setting, whereas subsequent papers are free to adapt it to particular applications and add bells and whistles. It is critical to keep the initial paper simple, so that the essence of the method is clearly conveyed, free from any add-ons that would make it unclear whether the method works at all without the add-ons, whether the method can be generalized to other domains (since add-ons typically cannot be) and whether performance gains are coming from the core method or the add-ons.

AnonReviewer3:

Comparing a given set of methods for training a particular model would not offer much conclusive evidence, because performance of generative models is sensitive to both the choice of model and the training method. If a model cannot accurately model the data, the relative performance of given methods does not say much, because the relative ranking of different methods may change on a different, better model. As a case in point, for Gaussian mixture models, E-M converges much more quickly than maximum likelihood. However, there are well-known examples where E-M converges extremely slowly. An experiment demonstrating the former would seem to suggest that E-M is better than maximum likelihood; an experiment demonstrating the latter would seem to suggest the E-M is worse. The truth is that E-M is sometimes better and sometimes worse. What we really care about is whether E-M works well on a model that we care about - because it does work well on a Gaussian mixture model, it has value. So we don’t agree that comparing various methods on simple models or Real-NVP would necessarily add much value to the paper.

AnonReviewer2:

First, we point out that our algorithm does *not* match the samples with their nearest data point; it matches each data point with their nearest sample. As explained in section 1.2, this is a subtle, but critical distinction: the former is similar to what a GAN with a nearest neighbour discriminator does and can collapse modes. Only the latter can be equivalent to maximum likelihood (because of the asymmetry of KL-divergence, as explained in our response to the comment below).

See our response to AnonReviewer1 regarding the particular nearest neighbour search algorithm that we used. Please note that the focus of this paper is not on the nearest neighbour search algorithm; please refer to the paper on the algorithm for performance evaluations.

Figure 2 shows the stability of training. The lack of vanishing gradients can be shown analytically, since the gradient of squared Euclidean distance can only vanish when the two points coincide exactly. It is difficult to empirically show the lack of mode collapse, since that would involve finding a way to compute recall, but doing so requires globally optimizing over the latent code, for which there is no efficient algorithm.

---

### Meta-Review · Area_Chair1 · 2018-12-16

**Confidence:** 4
**Recommendation:** Reject

**Metareview:**

The manuscript proposes a novel estimation technique for generative models based on fast nearest neighbors and inspired by maximum likelihood estimation. Overall, reviewers and AC agree that the general problem statement is timely and interesting, and the subject is of interest to the ICLR community

The reviewers and ACs note weakness in the evaluation of the proposed method. In particular, reviewers note that the Parzen-based log-likelihood estimate is known to be unreliable in high-dimensions. This makes a quantitative evaluation of the results challenging, thus other metrics should be evaluated. Reviewers also expressed concerns about the strengths of the baselines compared. Additional concerns are raised with regards to scalability which the authors address in the rebuttal.